# Breast Reconstruction: Economic Impact Swiss Health Insurance System

**DOI:** 10.3390/medicines9120064

**Published:** 2022-12-16

**Authors:** Jeanne Martin, Pietro G. di Summa, Wassim Raffoul, Nathalie Koch

**Affiliations:** Plastic Reconstructive and Aesthetics Surgery Department, Lausanne University Hospital, Rue du Bugnon 46, 1011 Lausanne, Switzerland

**Keywords:** breast cancer, reconstruction, cost, Switzerland

## Abstract

Background: Considering present concerns about healthcare costs and the lack of evidence and published articles on breast reconstruction costs in Switzerland, we retrospectively investigated charges to the Swiss healthcare system for different breast reconstruction procedures at the Centre Hospitalier Universitaire Vaudois. Methods: We selected all hospitalized patients at the University Hospital who underwent a “total” delayed breast reconstruction from January 2012 to December 2015. Analysis included 72 women who underwent autologous or implant-based reconstructions. Three main breast reconstruction techniques were included: Deep Inferior Epigastric Perforator (n = 46) autologous flap reconstruction, Tissue Expander followed by Implant (n = 12) and pedicled Latissimus Dorsi (n = 12) flap with or without tissue expander and implant (n = 7). For all different groups, the global costs of reconstruction and total number of required operations were statistically compared. Results: Global costs for Deep Inferior Epigastric Perforator reconstruction were 29,728 ± 1892 CHF (avg ± Std. Error of Mean), while Tissue Expander reconstruction showed a significantly higher global cost, reaching an average of 44,313 ± 5553 CHF (avg ± Std. Error of Mean). LD showed a similar cost, compared to the Deep Inferior Epigastric Perforator reconstruction (29,813 ± 3637 CHF), increasing when including an implant (37,688 ± 4840 CHF). No significant differences in the number of interventions were detected. Conclusion: These data show that autologous breast reconstruction (DIEP) delivers the best cost ratio, with lower overall costs. Implant-based reconstructions showed a greater likelihood of complications and re-intervention, globally creating superior costs when compared to autologous reconstructions.

## 1. Introduction

According to the Swiss Cancer League, more than 15 women develop breast cancer in Switzerland. This translates to approximately 5500 women a year [1] or 110 new cases per 100,000 inhabitants per year [2]. Breast cancer is, and has been for decades now, the deadliest cancer for women [2]. The mortality rate 5 years after diagnosis increases to 20% in women with a non-specified breast cancer stage at diagnosis [1]. Breast cancer is more frequently diagnosed in French-speaking Switzerland and in Ticino than in German-speaking Switzerland, where the mortality rate is higher [1].

Although almost 80% of the patients undergo a breast-conserving surgery [3], mastectomy remains an important part in the management of breast cancer. Some authors showed that nowadays, this option is more frequently chosen by patients than before, from 31% in 2003 to 43% in 2006 if women underwent an MRI before the surgery [4], especially in young women [5]. There is an increase in prophylactic mastectomies as well, partially due to the “Angelina Jolie Effect” [4], but mainly due to the greater use of genetic tests revealing mutations suggestive of breast cancer development, particularly in young women with a family history of breast cancer.

In France, where breast reconstruction (BR) is covered by health insurances, as it is in Switzerland, more than 80% of women with mastectomy choose to undergo a BR [5].

Different reconstructive procedures can be adopted to reconstruct breast volume and shape, generally divided in implant-based and autologous tissue reconstructions.

Among autologous tissue reconstructions options, DIEP flap is nowadays considered the gold standard for breast reconstruction [6], combining a significant reconstructive volume with reliability and preservation of the rectus abdomini muscles [7]. An LD flap may assure a shorter operative time with lower failure rates, thus, avoiding the need for microsurgical procedures. However, donor site morbidity is generally considered higher and reconstructive volume lower [8]. Implant-based reconstruction does not require an excess of abdominal tissue and the surgery is generally easier than the autologous options, while retaining the risks related to prosthetic implants, such as higher infections.

Further procedures, either on the reconstructed breast or on the contralateral, can be performed to improve the final breast contour and symmetry. The most performed are fat grafting to the reconstructed breast, contralateral side mastopexy-symmetrization and reconstruction of the nipple areola complex.

Reconstruction (using either implants or flaps) may be immediate or delayed after mastectomy. When adjuvant RT needs to be delivered, autologous reconstructions are generally preferred, as complications dramatically increase when prosthesis procedures are combined with radiotherapy [9].

When adjuvant RT is indicated, delayed reconstruction has been generally preferred in the recent past [10], even if immediate reconstruction followed by RT has progressively gained popularity [11]. 

In the complex scenario involving the timing of reconstruction and choice of procedures, costs have been widely addressed in the literature, but the final results are very different depending on the study criteria, where the health system of the country involved is likely to influence the overall cost of the reconstruction. In Switzerland, to the best of our knowledge, the health costs have not been studied yet, thus requiring further investigations, especially given the growing trend towards autologous reconstructions.

The aim of this study was to critically analyze costs on the Swiss healthcare insurance system, to point out eventual significant differences among different reconstructive procedures, and finally to define the most economic way of reconstruction.

This study focuses on three main types of delayed BR performed at the University Hospital of Lausanne (CHUV). Deep Inferior Epigastric Perforator (DIEP) flap, Tissue Expender followed by Implant (TE/I) and pedicled Latissimus Dorsi flap with or without tissue expander and implant (LD +/− TE/I). Immediate BR was excluded because immediate autologous breast reconstruction has been less performed between 2012 and 2015. The global costs of reconstruction, total number of required operations, total operation time and time to complete the whole reconstruction were statistically compared.

## 2. Patients and Methods

From January 2012 to December 2015, 80 patients were admitted for delayed breast reconstruction at the Reconstructive Surgery unit at the University Hospital of Lausanne (CHUV), Switzerland. Patients who underwent a total breast reconstruction by DIEP, LD +/− TE/I and TE/I, were included in this study, regardless of their ethnicity, age, stage at diagnosis, comorbidities or tobacco use. Bilateral and immediate reconstructions were excluded from this study because their limited number was insufficient to be statistically compared with other groups. By meeting the above criteria, 72 women were included in this retrospective analysis of breast reconstruction. Raw data on health costs were obtained according to the SwissDRG system (Swiss Diagnosis Related Group). The DRG system is used in Switzerland for the calculation of medical costs for inpatients. A cost-weight is attributed to each DRG, based on several criteria, such as the main diagnosis, secondary diagnosis, treatments, etc. The remuneration of each hospital in Swiss francs (CHF) is the result of the cost-weight multiplied by the base rate. The base rate, currently around CHF 10,000, is reviewed each year so that it can be adapted to be closer to the cost of stay [12]. Importantly, the duration of hospitalization does not affect the cost of stay, as the DRG is determined by the intervention and not by the number of nights spent in hospital.

The remuneration of ambulatory medical care is based on the TARMED system. In the same way as for DRG, each medical act corresponds to several points. The monetary value of the point is frequently re-evaluated by the cantonal authorities.

Using the hospital digital database, all medical details, from surgical procedure to complications and final outcomes, were collected. The medical encoding and archiving unit supplied information concerning the DRG and the cost-weight of each hospital stay. With these data, we were able to calculate the amount invoiced to the health insurance: cost-weight multiplied by the base rate. The base rate in 2012 and 2013 was CHF 10,400 and in 2014 and in 2015, it was CHF 10,350.

All costs deriving from patients’ reconstructive journey (including both DRG and TARMED systems) were considered, obtaining the total cost of each type of reconstruction. The number of required operations, the total operation time and the duration of the main surgeries were also collected and statistically compared among groups.

All the patients signed informed consent forms on the use of their data for research purposes after their admission.

The object of this study was outside the scope of the Swiss Research Legislation. Therefore, authorization from the ethics committee was not required for the publication of this study.

### Statistical Analysis

One-way analysis of variance (ANOVA) with the Bonferroni multiple comparison test was used to statistically compare groups (GraphPad Prism 7, software Inc, La Jolla, CA, USA), Significance was expressed as * *p* < 0.05; ** *p* < 0.01; *** *p* < 0.001.

## 3. Results

The study followed up 72 female patients over a two-and-a-half-year period after their first breast reconstruction intervention. A total of 63.89% (n = 46) of reconstructions were with the DIEP technique, 16.67% (n = 12) with TE/I, 9.72% (n = 7) with LD and 9.72% (n = 7) with LD + TE.

DIEP costs on average CHF 29,728 ± 1892 (ave ± SEM), while TE costs accounted for CHF 44,313 ± 5553 (ave ± SEM). The difference between these two groups was statistically significant (** *p* < 0.01). LD costs reached CHF 29,813 ± 3637 (ave ± SEM) and were increased when LD flaps were associated with an implant—CHF 37,688 ± 4840 (ave ± SEM).

The distribution of patient ages was homogenous, among groups, without statistically significant differences. Similarly, no difference in the number of procedures to complete the reconstruction was detected among implant-based and autologous reconstructions (Figure 1).

TE/I demand 3.72 ± 0.47 operations (ave ± SEM), while DIEP demand 2.97± 0.17 operations (ave ± SEM), LD 2.86 ± 3.67 (ave ± SEM) and LD + TE/I 3.57 ± 0.53 (ave ± SEM).

Concerning the total length of the reconstruction process, no significant difference was present among the interventions, ranging from 14.5 to 20 months. The operative time was significantly higher in DIEP reconstructions. The DIEP operative global time of reconstruction was significant longer when compared to TE/I (575 ± 30 min vs. 358 ± 90 min; ave ± SEM). No statistically significant differences were found when comparing with the other groups (LD 423 ± 45 min, LD + TE/I 416 ± 57 min, respectively; all expressed as ave ± SEM).

While a statistical analysis on the complications was impossible, considering the limited number of occurrences, the complications seemed more frequent during implant-based reconstructions than autologous reconstructions (Table 1). We also observed that DIEP autologous reconstructions required fewer revision surgeries than TE/I (17.4% et 33.3%, respectively). In the TE/I group, four women suffered from capsulitis, requiring revision surgery in the 2 years following their first intervention. One woman suffered from an infection in this group but received a conservative treatment. In the DIEP group, revision surgeries were conducted following three flap necroses, one of which had flap failure and another also had an associated infection, two hematomas and two thrombosis without damaging the flap. Another woman treated by DIEP suffered from partial necrosis, which could be treated conservatively without surgery. Regardless of the reconstruction, all the patients who suffered from an infection (n = 3/71) had a BMI greater than 30, though we were unable to find any significant association between infections and BMI.

The global complication rate detected in each group was for DIEP 19.6% (n = 9/46), TE/I 41.7% (n = 5/12), LD 8.3% (n = 1/12) and LD + TE/I 14.3% (n = 3/7)

## 4. Discussion

The present study shows significantly less expensive costs for DIEP reconstructions when compared to TE/I, which supports other articles published [13,14]. However, other European studies showed different conclusions: in Holland, the medical cost (including those caused by complications) of DIEP was EUR 17,351, EUR 9561 for LD + implant and EUR 15,690 for TE/I [15]. In the USA, the total cost (including complications) of DIEP reached USD 23,120.49 and USD 22,739.91 for TE/I [16], while in the UK, excluding the cost of complications, DIEP costs GBP 9144, LD + implant GBP 6654 and GBP TE/I 3427 [17]. Unsurprisingly, due to the high cost of living and general pricing, these interventions are costlier in Switzerland than in other European countries [15,16,17]. Healthcare costs showed that DIEP reconstructions were significantly less expensive that TE/I. LD + TE/I also seems to be less expensive than TE/I alone, probably because of a lower overall complication rate. No other significant differences in costs among the other reconstructions were noticed.

From analyzing the data, we could detect that the first surgical step in the DIEP intervention (free flap transfer) costs more than the first TE/I (expander insertion). However, this value rapidly inverts in the secondary touch-up procedures, which for DIEP, are generally minor (lipofilling, liposuction or contralateral breast reduction, symmetrizing mastopexy, mammoplasty augmentation), whereas the second intervention in the TE/I group is more extensive due to the withdrawal of the expander and the insertion of the permanent prosthesis. Furthermore, women that have undergone a reconstruction with an implant may require an additional surgical procedure 10–15 years after the intervention to replace the implant. Importantly, this was not taken into account in our study, but it needs to be underlined as it would further increase health costs in the medium and long term. In fact, prosthesis changes imply supplementary interventions over time, potential surgical risks, further paramedical costs for patients and social consequences (work leave, travel expenses).

The main reason for implant exchange is due to capsular contracture. This affects 2.8 to 45% of women [18,19,20,21], meaning that 0.2 to 5.4 women of our study group would be expected to develop capsular contracture, resulting in pain, discomfort and potential further surgical treatments. Published research does not agree on when contracture occurs: some studies suggest it occurs relatively early, with 92% of cases arising in the first year post-operation [18], while other studies suggest an incidence curve plateauing at 8 years after the intervention [22].

Implant shifts and malrotations are another complication of procedures involving prostheses that can require further surgery. This further increases the total cost of this type of reconstruction. In our study, 33.3% of patients with an implant had to have revision surgery due to complications, which is a similar figure to previous research [21]. Since the DRG system was only introduced to Switzerland in 2010, medium- or long-term costs could not be estimated in this study. Additionally, this study did not consider the paramedical costs associated with reconstruction, such as transport costs, quality of life and work leave. The cost of absence from work could strongly influence the total cost.

In the literature, the rate of secondary surgery for complications is higher in TE/I than autologous breast reconstruction interventions [13,14,20,21]. Our data seem to confirm these reports and may explain the observed differences in costs. Aesthetic outcomes were not evaluated in present study. However, particularly when radiotherapy is used, autologous reconstructions can guarantee a better breast contour and tissue quality [9].

The operative time was significantly greater for DIEP compared to TE/I reconstructions. By involving free tissue transfer and microsurgical time, DIEP took significantly longer than the techniques involving the fitting of prosthesis due to the first intervention being more complex than it is with TE/I.

The number of interventions accounting for a complete reconstruction and global reconstruction time did not differ among groups. Indeed, after the first intervention, there was a waiting time of 3 to 6 months before the symmetrization procedure, and a further 3 months before the nipple reconstruction.

The risk factors we explored were pre-surgical radiotherapy, smoking and diabetes, as well as BMI. Considering the small sample size of the study, no significant influences of such risk factors on breast reconstruction outcomes were detected.

This study shows lower complication rates in DIEP reconstructions when compared to implant-based reconstructions. This goes in line with the worldwide trend, where DIEP reconstructions are progressively becoming a routine procedure in major breast units.

All the aforementioned points suggest that DIEP should be the most suitable intervention as far as cost-effectiveness is concerned. However, it is relatively contraindicated in case of smoking, diabetes, high cholesterol and previous laparotomies. Moreover, body habitus need to be considered, making this procedure impossible in underweight women, where other autologous reconstruction options may be more appropriate. We used LD as an alternative for patients with contraindication to abdominal flap surgery or for smokers. The main drawback was a limited breast volume, occasionally requiring an implant.

This study presents different limitations which need to be acknowledged. Firstly, there was a limited number of patients involved and there were relative differences in patient numbers among groups. This was mainly due to the fact that the DRG system was recently introduced, thus limiting the number of patients to be included. Indeed, less frequent procedures such as bilateral reconstructions and immediate breast reconstructions with implants were not included or analyzed as they were insufficient in number to allow for valid statistical comparisons. Moreover, it is likely that in the space of a few years, a larger sample will allow for new comparisons to become significantly meaningful. The same concept may count for the influence of supposed risk factors (tobacco, radiotherapy and obesity) on complications.

## 5. Conclusions

Autologous breast reconstruction (DIEP) shows the best cost–benefit ratio, with lower overall costs. Implant-based reconstructions show a greater likelihood of complications and re-intervention, globally creating superior costs when compared to autologous reconstructions.

Therefore, breast reconstruction with a DIEP flap is the most efficient technique and it should be taken into consideration in patient management. Nevertheless, the choice of the reconstruction technique remains with the patient.

## Figures and Tables

**Figure 1 medicines-09-00064-f001:**
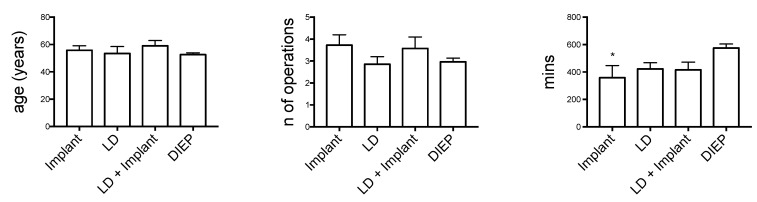
Analyzed data. * *p* < 0.05; ** *p* < 0.01.

**Table 1 medicines-09-00064-t001:** Complications.

Complications	Implant (n = 12)		LD (n = 7)		LD + Implant (n = 7)		Diep (n = 46)		Total (n = 72)	
	n	(%)	n	(%)	n	(%)	n	(%)	n	(%)
Infection	1	(8.33)	0	(0.00)	0	(0.00)	2	(4.35)	3	(4.17)
Hematoma	0	(0.00)	0	(0.00)	1	(14.29)	2	(4.35)	3	(4.17)
Capsulitis	4	(33.33)	0	(0.00)	1	(14.29)	0	(0.00)	5	(6.94)
Seroma	0	(0.00)	0	(0.00)	1	(14.29)	0	(0.00)	1	(1.39)
Necrosis	0	(0.00)	0	(0.00)	0	(0.00)	4	(8.70)	4	(5.56)
Thrombosis	0	(0.00)	0	(0.00)	0	(0.00)	2	(4.35)	2	(2.78)
Reoperation	4	(33.33)	1	(14.29)	1	(14.29)	8	(17.39)	14	(19.44)

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
