# Peer review of "Breast Reconstruction: Economic Impact Swiss Health Insurance System"

_medicines, 2022, doi:10.3390/medicines9120064_

Round 1
Reviewer 1 Report
This is a study aimed to compare the costs of several types of delayed breast reconstruction surgeries for breast cancer patients in Switzerland. The topic is important and interesting. There are some issues that must be addressed before publishing-
- The introduction should be much more precise and focused- it’s role is to briefly present the problem or the question to which this study proclaims to answer.
o The paragraph about screening in the introduction is redundant and not relevant to the topic. I suggest removing it.
o Also, the paragraph about indications for mastectomy is not relevant to my opinion. Since this paper deals with cost of breast reconstruction after mastectomy, the reason to undergo mastectomy does not matter.
o I suggest focusing the introduction on relevant issues such as: in what aspects delayed reconstruction is different from immediate in terms of costs? what is the percentage of delayed reconstruction out of all breast reconstructions? What is the current literature on the cost of delayed reconstruction?
- The authors should better explain why immediate reconstruction patients were excluded from the study? If the primary goal is to examine only the delayed-reconstruction patients (perhaps because costs are sharply different?) than why it says that immediate-recon pts were excluded d/t their limited number? Most breast cancer patient undergo immediate rather than delayed reconstruction, so that including them in the analysis should make the power of this study much higher. (the biggest limitation here is the small sample size)
- ‘The mortality rate 5 years after diagnosis rises as high as 20% in women with non-specified breast cancer stage at diagnosis [1]’- sounds way too high to me, but the reference is not in English.
- ‘DIEP flap is nowadays considered the gold standard’- by whom? Is it in formal guidelines?
- How does the authors explain the fact that TE/I was more expensive than LD+TE/I?
- In such small numbers it would be reasonable to mention the absolute number in the ‘Results’ section rather than the percentage only.
- ‘women that have undergone a reconstruction with an implant generally require an additional surgical procedure 10-15 years after the intervention to replace the implant.’- this is an important note, however I would be more careful in the phrasing ‘generally require’ since it is not commonly done for most patients nowadays, so I suggest changing it to ‘may require’.
- The ‘Aesthetic outcomes’ issue is mentioned twice in the ‘Discussion’- remove one of them (or even both, it’s not the question of the study).
-
- I suggest wrapping up the paper in a short conclusion paragraph.
Author Response
Dear Reviewer
The English language has been improved.
The introduction has been changed and paragraphs have been removed.
An explanation has been given about the immediate reconstruction.
The reference about DIEP ad a gold standard has been changed.
An explanation about the different cost between TE/I and LD-TE/I has been given
The absolute number in the results have been mentioned.
The sentence has been changed to : may required.
The aesthetic outcome has been removed.
A conclusion has been written

Reviewer 2 Report
This is obviously a paper relevant very locally to Switzerland which would appear to have a different costing system as the other countries quoted in the discussion. As the DRG system does not take into account the length of stay I think a comment about that between free flap patients and expander/Implant patient would be appropriate, as is the time away from work both of these would add to the cost of the reconstruction albeit not being able to be accounted for accurrately.
An important paper for policy makers in Switzerland but there are so many vaiables for a patient undergoing breast reconstruction I believe they have a right to choose the method of breast reconstruction within various constraints. eg: What would women choose if given a choice in breast reconstruction. Lam et al PRSGO 2016 Sep; 4(9): e1062.
Author Response
Dear Reviewer
The English has been improved.
A comment about the time away from work has been introduced.
A sentence has been introduced about the right to choose the reconstruction for the women
